Learning-based short text compression using BERT models

Öztürk Emir emirozturk@trakya.edu.tr
Mesut Altan
Department of Computer Engineering, Trakya University , Edirne , Turkey
Cirillo Stefano
Electronic publication date: 2024 Oct 18
Publication date: 2024
Volume: 10
Electronic Location ID: e2423
Received 2024 May 29; Accepted 2024 Sep 25
Copyright: ©2024 Öztürk and Mesut
Copyright year: 2024
Copyright holder: Öztürk and Mesut
License: This is an open access article distributed under the terms of the Creative Commons Attribution License, which permits unrestricted use, distribution, reproduction and adaptation in any medium and for any purpose provided that it is properly attributed. For attribution, the original author(s), title, publication source (PeerJ Computer Science) and either DOI or URL of the article must be cited.
License URL: https://creativecommons.org/licenses/by/4.0/

Keywords: BERT, Fine tuning, Learning-based compression, Text compression

Funding: The authors received no funding for this work.

==============================
Learning-based data compression methods have gained significant attention in recent years. Although these methods achieve higher compression ratios compared to traditional techniques, their slow processing times make them less suitable for compressing large datasets, and they are generally more effective for short texts rather than longer ones. In this study, MLMCompress, a word-based text compression method that can utilize any BERT masked language model is introduced. The performance of MLMCompress is evaluated using four BERT models: two large models and two smaller models referred to as “tiny”. The large models are used without training, while the smaller models are fine-tuned. The results indicate that MLMCompress, when using the best-performing model, achieved 3838% higher compression ratios for English text and 42% higher compression ratios for multilingual text compared to NNCP, another learning-based method. Although the method does not yield better results than GPTZip, which has been developed in recent years, it achieves comparable outcomes while being up to 35 times faster in the worst-case scenario. Additionally, it demonstrated a 20% improvement in compression speed and a 180% improvement in decompression speed in the best case. Furthermore, MLMCompress outperforms traditional compression methods like Gzip and specialized short text compression methods such as Smaz and Shoco, particularly in compressing short texts, even when using smaller models.

Introduction

Short text compression is crucial for optimizing storage and improving the transmission efficiency of small data packets, especially in environments with limited bandwidth or storage capacity. In certain cases, data can be combined and compressed to increase the compression ratio. However, there are also scenarios where merging and compressing data is not possible. For example, real-time applications such as live chat systems or streaming services, or SMS messages, require data to be processed and transmitted as soon as it is created and do not allow for data aggregation.

High compression ratios could not be achieved often when short texts are compressed with traditional data compression methods. If the size of the text is too small, there may be expansion rather than compression. This is because short texts contain few repeating substrings (words or n-grams). The reason why short texts cannot be compressed well with a data compression method using a semi-static or dynamic model is that a dictionary containing these substrings must also be sent to the decoder with the compressed text. The dictionary created for a large text is much smaller than the compressed size of that text, while for short texts the dictionary size can be larger than the compressed text data.

One method of being able to highly compress short texts is to use static dictionaries on both the encoder and the decoder. In this case, the compression ratio will increase as the dictionary will not need to be transferred. The disadvantage of using a static dictionary is that only texts compatible with the dictionary used can be compressed well. For example, a word-based static dictionary for English texts will mostly consist of stopwords such as ‘the’, ‘in’ and ‘of’ and will not be able to compress short texts in different languages. In order to achieve good compression ratios for texts in different languages, it will be necessary to have different static dictionaries for different languages on both the encoder and decoder, determine the language of the text to be compressed and compress the text with the corresponding dictionary. There are some studies that use machine learning to create these different static dictionaries.

Standard compression methods often struggle to effectively compress short texts because they either store the dictionaries they generate or rely on fixed dictionaries. In many cases, the generated dictionaries or the extracted statistics that need to be stored take up more space than the text itself. Additionally, with fixed dictionaries, the prediction accuracy declines due to the limited word space, making it difficult to achieve good compression ratios across different files. Therefore, it is evident that improving the probabilistic prediction accuracy of words is necessary. Considering the challenges in compressing short texts, it is necessary to derive statistics from more generalized data to enhance compression ratios. This can be achieved by training artificial intelligence models capable of word prediction using much larger datasets, which falls under the category of learning-based methods. Until recent years, data compression methods based on long short-term memory (LSTM) architecture, which is a type of recurrent neural network (RNN), are commonly employed to achieve high compression ratios. However, the use of transformers for this purpose has gained popularity in the last few years.

Although methods like LSTM and RNN are small and fast, their performance is limited because they are trained from scratch. Consequently, they are unable to achieve results beyond a certain level of effectiveness. To overcome these limitations, transformers, which are pre-trained, capable of bidirectional training, and offer higher performance, can be utilized.

The transformer architecture is introduced by Google Brain in 2017 and has since become one of the most preferred NLP approaches for translation, text summarization, classification, question answering, and text generation. Unlike RNNs, transformers do not necessarily process data sequentially. For example, in a natural language sentence, the transformer does not need to process the beginning of the sentence before its end. Instead, it defines the context that gives meaning to each word in the sentence. This feature allows better parallelization than RNNs and therefore reduces training times (Vaswani et al., 2017).

The most well-known Transformer-based pretrained systems are Bidirectional Encoder Representations from Transformers (BERT) developed by Google (Devlin et al., 2018) and Generative Pre-trained Transformer (GPT) developed by OpenAI (Radford et al., 2018). Although there are studies with BERT in many different areas such as sentiment classification (Gao et al., 2019), neural machine translation (NMT) (Clinchant, Jung & Nikoulina, 2019), cross-lingual information retrieval (CLIR) (Litschko et al., 2022) and question answering systems (Kazemi et al., 2023), text compression with BERT is an understudied research area.

In this study, we propose the MLMCompress method, which leverages BERT-based models for word prediction to achieve data compression. MLMCompress is capable of utilizing any model that performs word prediction. The study employs pre-trained BERT models with the aim of enhancing word prediction accuracy and, consequently, improving compression ratios. Two large general-purpose BERT models and two smaller BERT models are selected and utilized to obtain experimental results. The large models, being already trained and generalized, are not finetuned to ensure their broad applicability, as discussed in the discussion section. In contrast, the smaller models initially exhibited lower performance due to their limited number of parameters and are fine-tuned to explore the outcomes of customizable scenarios. To ensure fairness in the results, the training and testing datasets are prepared as two independent datasets.

The next section of this article reviews previous work on learning-based compression and short text compression. The proposed method is detailed in the third section, followed by the presentation of experimental results in the fourth section. The ‘Discussion’ section outlines the purpose, findings, and limitations of the method, while the final section provides the conclusions.

Related Work

In the 1990s, methods based on the Prediction by Partial Matching (PPM) algorithm (Cleary & Witten, 1984) gave the best compression ratios in many compression tests. The PAQ data compression method, which is developed by Matt Mahoney in the early 2000s (Mahoney, 2002), started to use artificial neural network (ANN) in the model mixer stage of the PAQ7 version released in 2005 (Mahoney, 2005). With the additional improvements made by many different researchers on the PAQ8 version released in 2006, the highest compression ratios are achieved in data compression competitions such as the Hutter Prize and the Calgary Compression Challenge. Although the PAQ8 compresses and decompresses quite slowly, it is faster than previous compression methods using ANN. PAQ8 has inspired the increasing use of neural networks in data compression (Knoll & de Freitas, 2012).

Learning based compression

In recent years, many learning-based data compression methods have been developed that work much slower than traditional compression methods but can offer higher compression ratios.

Cmix, one of the learning-based data compression methods developed by Byron Knoll, has three different stages. The preprocessing stage transforms the input data into a more compressible form. The model prediction stage uses more than 2,000 independent models specialized for certain data types such as text, executable files or images. The context mixing stage includes the byte level LSTM mixer and the bit level context mixer similar to the PAQ8. The output of the context mixer is refined using SSE (secondary symbol estimation) algorithm. Knoll created two other compressors using only LSTM: lstm-compress and tensorflow-compress.

NNCP, developed by Fabrice Bellard, is a lossless data compression method that uses neural network models based on LSTM and Transformer architectures (Bellard, 2019). In the tests of the first version, LSTM gave better results than Transformer. In the second version developed two years later, Bellard used only the Transformer architecture and significantly improved its performance (Bellard, 2021). Bellard also developed a program called gpt2tc, which compresses English texts or performs auto-complete on them using the GPT-2 model.

In recent years, there have been significant advancements in utilizing transformer-based models and large language models (LLMs) for various applications. For instance, a transformer-based method, GPTZip, performs compression using the GPT-2 model (Nesse & Li, 2023). In this study, to assess its effectiveness on shorter or multilingual texts, the GPTZip is integrated into experimental evaluations.

Sentence compression is the process of producing a shorter sentence by removing some unnecessary words while preserving the grammar and important content of the original sentence. Sentence compression is actually a type of lossy text compression. Although there are some sentence compression studies using BERT (Niu, Xiong & Socher, 2019; Nguyen et al., 2020; Park et al., 2021; Jun, 2020), no study has been found in the literature in the field of lossless text compression.

Short text compression

There are simple and fast methods developed to compress short texts, most of which just use a static dictionary. Smaz is a simple method developed by Salvatore Sanfilippo that can compress short texts quickly and at good ratios. It compresses English text better than other languages, as it contains a static dictionary specific to English text written in lowercase letters. It can also compress HTML and URLs, as the dictionary also includes entries such as “http://” and “ </”.

Shoco is a method written in C language by Christian Schramm to compress short texts. The default compression model is optimized for English words, but another compression model can be created with different input data. Shoco is an entropy encoder and generally gives worse compression ratios than dictionary-based Smaz when compressing English texts. However, if there are many words in the text that are not included in the dictionary, such as numbers, Smaz can expand the text. Shoco, on the other hand, never expands the size of ASCII type texts (it can increase non-ASCII characters by two times).

The Short Message Arithmetic Compressor (SMAC) uses a word-based static dictionary, 3rd order letter statistics and arithmetic coding to compress short text messages such as SMS and Twitter (Gardner-Stephen et al., 2013). It is slower than Shoco and Smaz; however, it provides higher compression ratio. It offers full Unicode support and accepts UTF-8 strings. SMAC never increases the length of a string during compression, and uncompressed strings can be passed to the decompressor.

b64pack is another algorithm developed for the compression of short text messages (Kalajdzic, Ali & Patel, 2015). In the first stage of the algorithm, the input text is converted into a format that can be well compressed in the second stage. The second stage consists of a transformation that reduces the size of the message by a fixed fraction of its original size. b64pack is a simple algorithm that can only be used for English texts. It is fast but offers a moderate compression ratio.

In Aslanyürek & Mesut (2023), static dictionaries are created with machine learning for different languages and different topics. The compression method determines which static dictionary is more compatible with the text to be compressed and compresses it with that dictionary (Aslanyürek & Mesut, 2023). Although this method is slower than the others, better compression ratios can be obtained due to the use of different static dictionaries.

Word based compression

Word-based text compression methods are techniques where symbols are selected as words. By substituting frequently occurring words with shorter symbols or codes, the overall size of the text can be reduced. This approach is particularly useful in applications where efficient search and retrieval of information are crucial, such as in databases and information retrieval systems. While these methods may not achieve the highest possible compression ratios, they offer a balanced trade-off between compression efficiency and the ease of performing operations like searching and indexing on the compressed text. In these methods, the symbol substitution process is applied to words, and sometimes the primary goal is not maximum compression but rather indexing and searching within the compressed text.

End-Tagged Dense Code (ETDC) and (s,c)-Dense Code (SCDC) are word-based semi-static compression methods that are effective when searching for a single word in a compressed text (Brisaboa et al., 2007). Dynamic ETDC (DETDC) and Dynamic SCDC (DSCDC) are better than semi-static ones in terms of compression speed, but they are worse in terms of decompression speed (Brisaboa et al., 2008). Dynamic Lightweight ETDC (DLETDC) and Dynamic Lightweight SCDC (DLSCDC) are able to perform decompression as fast as semi-static dense codes (Brisaboa et al., 2010). Although the compression speeds of dynamic lightweight codes are worse than dynamic dense codes, they are better than semi-static dense codes. On the other hand, dynamic lightweight dense codes are not better than dynamic dense codes, and dynamic dense codes are not better than semi-static dense codes in terms of compression ratio.

The Tagged Word-Based Compression Algorithm (TWBCA) is another word-based semi-static compression method that allows searching for a word or a phrase in compressed text (Buluş, Carus & Mesut, 2017). Although TWBCA is slow at decompression, it is better than ETDC in compression speed.

Multi-stream word-based compression algorithm (MWCA) is a method similar to ETDC, but differs from ETDC in that it produces multi-stream output (Öztürk, Mesut & Diri, 2018). Exact word matching can also be done on compressed texts with MWCA. Although the compression ratio and decompression speed of MWCA are worse than ETDC, the compression speed is similar to ETDC, and the multi-stream output structure of MWCA allows much faster compressed search than ETDC.

MLMCompress

MLMCompress operates on the principle of taking fixed-size windows from the text and predicting the next word based on these windows. Compression is achieved through encoding when the predicted words match those in the text. Any trained model capable of producing a word output for given a specific input can be used for word prediction. In this study, BERT models, which offer a balanced trade-off between speed and performance, are employed for word prediction.

BERT models can be used for masked language modelling (MLM). In the MLM method, the word given with the mask tag is predicted by BERT. For example, it suggests “rock” for an input such as “very heavy [MASK]”, while it suggests “man” for an input such as “very strong [MASK]”. Better prediction results are obtained because the masked word is predicted according to the words before and after it. The maximum number of words to be obtained as a result of the prediction can also be given as a parameter to BERT. The number of words obtained may not always reach this maximum value.

MLMCompress compresses text using this prediction mechanism of BERT. Although BERT can work bidirectionally (using both previous and next words) for masked word prediction, MLMCompress only looks at previous words. This is because the decompressor would not have any information about the next word during the decompression process.

The compression and decompression stages of MLMCompress is given in Fig. 1.

The method creates a list of pairs ([Window, [MASK]]) by taking a window-sized segment of the obtained words and the corresponding next word, shifting one word at a time throughout the text. The aim here is to pre-process all windows and the words that will predicted, and then feed them into the model as a batch for prediction. This approach eliminates the overhead of model prediction for each word, thereby reducing the compression time. Any model capable of taking a fixed-size input and suggesting a desired number of words as output can be used for prediction. In this study, BERT models are selected for this purpose. One advantage of the BERT model is its ability to suggest a specified number of words for a given window. Increasing the number of suggested words improves the likelihood of BERT predicting the same word that appears in the text, although this also slows down the prediction process.

After all windows and predicted word results are obtained, each window and its corresponding word are checked sequentially. At each stage, it is verified whether the word in the text is predicted by BERT. If the word is found in BERT’s predictions, compression is achieved by using the position of BERT’s prediction for the encoding process. If the word cannot be predicted by BERT, it is added to the R sequence. The R sequence is then encoded using Huffman coding, and the compressed file is created.

During the decompression phase, batch processing is not possible. First, a window is created using R. Then, using the window and the code sequence, the decompression process is carried out. Each window is sequentially fed to the BERT model, and if an index was encoded during the compression process, BERT’s prediction at that position is obtained as the word. If the code is 0, a word from the R is added; if the code is 1, a space is added. This escape code is applied specifically when there are multiple spaces between two words, rather than a single space. Therefore, to prevent BERT from predicting spaces and to improve performance, the value of 1 is used as an escape code to represent these spaces. For index values greater than 1, the count of escape sequences, which is 2, is subtracted from the index, and the word corresponding to this new index is selected from the BERT predictions. In the final stage, the output file is produced.

Figure 1 Compression and decompression stages of MLMCompress.

The dataset and models used for compression are described in the next subsections.

Dataset used for compression

For fine-tuning the small models, a 20 MB of text is extracted from the English 50 MB file which is in the Pizza&Chili corpus, and sentences are obtained from this subset. Subsequently, datasets are obtained from Huggingface for six different languages, 20 MB of text is extracted from each of them to create multilingual training data. Datasets taken from Huggingface are named as “germanquad” (Möller, Risch & Pietsch, 2021), “Spanish speech text”, “xnli2.0 train French”, “Italian tweets 500k”, “Dutch social” and “Turkish instructions” respectively. Emojis, punctuations and links are removed from texts and obtained data is used for training.

The datasets consist of raw text rather than structured content. To enhance the prediction accuracy of the models in the desired language, whether in English or multilingual, raw texts in natural language are collected and used for fine-tuning. At this stage, apart from the tokenization performed by the BERT models, no additional feature selection or extraction is applied.

The obtained sentences are used for training the selected TinyBERT models. Before training, the dataset for multilingual and English data is combined and shuffled. During the training process, to prevent possible overfitting, the dataset is divided into training and validation sets as 90%–10%, and the loss values for both cases are obtained. The datasets used, along with their sizes and training-validation ratios, are provided in Table 1.

Table 1 The details of the datasets used for training and validation.

Language	Dataset	Size	Selected size	Train size	Validation size	
de	germanquad	20 MB	20 MB	18 MB	2 MB	
en	Pizza&Chili corpus	50 MB	20 MB	18 MB	2 MB	
es	Spanish speech text	38 MB	20 MB	18 MB	2 MB	
fr	xnli2.0 train French	82 MB	20 MB	18 MB	2 MB	
it	Italian tweets 500k	27 MB	20 MB	18 MB	2 MB	
nl	dutch social	25 MB	20 MB	18 MB	2 MB	
tr	turkish instructions	21 MB	20 MB	18 MB	2 MB	
Total			140 MB	126 MB	14 MB	

Models used for compression

MLMCompress allows the use of any masked language model for compression. Compression ratio of MLMCompress is highly correlated with prediction accuracy of masked language model. There are several options available for improving prediction accuracy. One of them is to use a larger model, but this could result in slower compression due to slower word prediction phase. Another option is to enhance prediction performance by fine-tuning a smaller model, thus achieving higher compression and decompression speeds.

There are fine-tuned versions of BERT models for different languages in Huggingface. These models are available as cased and uncased. In this study, the following four different cased models, two of which are large and the other two are small, are used:

• Model A: bert-base-cased / Large / English

• Model B: bert-base-multilingual-cased / Large / Multilingual

• Model C: tzyLee/quant-tinybert / Small / English

• Model D: dbmdz/bert-tiny-historic-multilingual-cased / Small / Multilingual

While large-sized models are used with MLMCompress without any preprocessing, the small-sized models are fine-tuned and then used in tests. The training parameters of the models are provided in Table 2.

Table 2 Training parameter values used in experiments.

Parameter	Value	
Epochs	100	
Max length	50	
Whole word mask	True	
Mask probability	0.15	
Learning rate	0.0003	

The models are trained on the dataset for 100 epochs. Since the models are designed for short text compression, the maximum token length is set to 50. Given that compression will be performed in different languages, and especially in agglutinative languages where different suffixes necessitate the suggestion of different words, the whole word mask parameter is set to true. The mask probability parameter, which indicates the likelihood of a word being masked in a given sentence during training, is left at its default value. The training and validation loss values obtained during the training process are presented in Fig. 2.

Figure 2 Train and validation loss values for tzyLee/quant-tinybert (A) and dbmdz/bert-tiny-historic-multilingual-cased (B).

As shown in Fig. 2, the variation in the loss values of the models is very small between 80-100 epochs. Therefore, the training is concluded in 100 epochs. In the final state, the loss value for tzyLee/quant-tinybert is obtained as 0.081, and the validation loss value is 0.06. For the dbmdz/bert-tiny-historic-multilingual-cased model, the loss and validation loss values are obtained as 0.7638 and 0.7620, respectively.

The compression and decompression methods using this model are described in the following subsections, respectively.

Compression method

MLMCompress starts the compression process using the maximum window size and prediction count parameters. Maximum Window Size refers to an array of words of a size that will be given to BERT for making predictions. Since the compressed word is encoded as one byte and two escape characters are required for uncompressed words and empty strings obtained at input splitting process, a maximum of 254 is used as the prediction count parameter.

In the first step of compression, the input is split into words W[w0, w1, …, wm]. At first, w0 is added to the R array for proper decompression. After this step, starting from w0 each window for prediction is prepared with a [MASK] token in the end and added to an array named I. Respective words will be added to the window until the window size reaches maximum window size (n). When the window size reaches n, the window update process is performed by removing the first word from the window and adding the next word to the window at each stage. The reason for using an array like I instead of making predictions in the first place is to reduce the time cost of the prediction phase of BERT. BERT can make predictions for batch inputs as a whole and efficiency is gained by pre-creating all windows to be predicted and providing them to BERT in a batched manner.

After obtaining Predictions from BERT, r predictions are requested for Window[PC] as Predictions[PC] {p1, p2, …, pr}. If the masked word exists in the prediction list, its index value is written into the C array. If the word is not in the Predictions[PC], the escape character ‘0’ is written into the C array and word is written into the R array.

After the creation of the arrays is completed, the R array containing words is converted into a string by combining space characters between the words and is encoded with Static Huffman Code Tables. If a small text is tried to be compressed with the Dynamic Huffman method, expansion will occur rather than compression, since the coding table obtained from the Huffman tree must also be sent to the decoder. For this reason, Static Huffman Code Tables are created for seven different languages used in the tests and these tables are used in both encoding and decoding. Finally, the C and Huffman encoded R arrays are concatenated and written to the output (compressed) stream. The steps of the compression process are given in Algorithm 1 .

_________________________________________________________________________ Algorithm 1 Compression algorithm of MLMCompress   1:  W = split input text by spaces and punctuations   2:  Window = [W[0]]   3:  R = [W[0]], C = [], I = []   4:  for i=1 to Length(W) do  5:     if W[i] is not empty then  6:         Add a list [Window, [MASK]] to I  7:         Add new word to Window  8:         if Length(Window) = n {Max window size} then  9:            Remove Window[0] 10:         end if 11:     end if 12:  end for 13:  r as prediction size 14:  Predictions = BERTPredsForAllWindows(I,r) 15:  PC = 0 as prediction counter 16:  for i=1 to Length(W) do 17:     if W[i] is empty then 18:         Append 1 byte to C 19:     else 20:         if W[i] is in Predictions[PC] then 21:            Ind = index of W[i] in Predictions 22:            Append Ind+2 to C 23:         else 24:            Append 0 to C 25:            Append W[i] to R 26:         end if 27:         PC = PC + 1 28:     end if 29:  end for 30:  return  Concat(C, Huffman(R)) _______________________________________________________

An example for compression of the sentence “He feels better since it is a sunny day” is given in Table 3. It is assumed that BERT makes 3 predictions (r = 3) and the maximum window size is 2 (n = 2). In the first step, the word “He” is given to BERT and predictions are taken from it. Since BERT predicts the word “feels” as the 3rd word, the value ‘3’ is coded into the C array and the window is updated by adding the word “feels”. In this step, the first word “He” is also written into R array for decompression. In the next step, the window “He feels” is given to BERT and the predictions are received. Since BERT cannot guess the word “good”, a ‘0’ byte is added to the C array and the word “better” is written to the R array. Since the window size is 2, the word “He” is removed from the window and the word “better” is added to the window after this stage. Since the word “since” is predicted as the 1st word by BERT, the index 1 is written to the C array. The process is applied to all remaining words using these three rules. After the R array has gone through the Huffman coding process, the C and R strings are combined and written to the output stream.

Table 3 Compression steps of sentence “He feels better since it is a sunny day”.

Window	Next word	BERT predictions	C	R	
He	feels	is, looks, feels	3	He	
He feels	better	good, bad, strange	3, 0	He, better	
feels better	since	since, here, too	3, 0, 1	He, better	
better since	it	today, that, then	3, 0, 1, 0	He, better, it	
since it	is	was, is, knows	3, 0, 1, 0, 2	He, better, it	
it is	a	what, known, a	3, 0, 1, 0, 2, 3	He, better, it	
is a	sunny	book, dog, pencil	3, 0, 1, 0, 2, 3, 0	He, better, it, sunny	
a sunny	day	day, park, side	3, 0, 1, 0, 2, 3, 0, 1	He, better, it, sunny	

Decompression method

In the first step of decompression, the C and Huffman encoded R arrays are separated and the unencoded R array is obtained by Huffman decoding. After this step, the first word is taken from the R array and the index values are read from the C array. At each stage, if the next index value is not ‘0’, prediction is performed by giving the window and r to BERT. After the predicted words are taken from BERT, the word from Predictions array at the index value which is read from the C array is added to the output stream. If the index value is ‘0’, the next word from the R array is added to the output stream. After this step, if the window size is less than n, this word is also added to the W. If the window size is equal to n, the first word in the window is removed and the last word is added to the end. When the process is complete, the output (decompressed) stream is obtained. The creation processes of the output stream are given in the Algorithm 2 .

__________________________________________________________________________ Algorithm 2 Decompression algorithm of MLMCompress   1:  C, HuffR = Split(Compressed)   2:  R = HuffmanDecode(HuffR)   3:  D = R[0] + ” ”   4:  W = [R[0]]   5:  RCounter = 1   6:  CCounter = 0   7:  while CCounter < Len(I) do  8:     if C[CCounter] = 0 then  9:         Append R[RCounter] + ” ” to D 10:         Refresh Window with adding R[RCounter] 11:         RCounter = RCounter + 1 12:     else if C[CCounter] = 1 then 13:         Append a space to D 14:     else 15:         Predictions = BERTPredsForWindow(W) 16:         Word = Predictions[C[CCounter]-2] 17:         Append Word + ” ” to D 18:         Refresh Window with adding Word 19:     end if 20:     CCounter = CCounter + 1 21:  end while 22:  return  D _____________________________________________________________________________________

An example for the decompression method is given in Table 4. To facilitate easy tracking of the next element, the processed elements are removed from the table rows. In the algorithm’s operation, these values are read by incrementing the CCounter value over the list. After the C and R arrays are obtained, the first word in the R array “He” is written into the window and output stream and predictions are taken from BERT. Since the value ‘3’ is read from the C array, the 3rd prediction is taken from the prediction list and written to the output stream. After the window is updated to “He feels”, CCounter will be 1 and the next value ‘0’ is read from the C array. Since the value ‘0’ means that BERT could not predict the word, the next word from R is written to the output stream without using prediction procedure. Since the window size is 2 for this example, the word “He” is removed from the window and the word “better” is added to the window. In the next step, since the last value in C is “1”, the 1st BERT prediction made for the “feels better” window, the word “since”, is written to the output stream. The process is applied to the end of C and R arrays and the decompressed stream is obtained.

Table 4 Decompression steps of sentence “He feels better since it is a sunny day”.

C	R	Window	BERT predictions	Output	
3,0,1,0,2,3,0,1	He, better, it, sunny	He	is, looks, feels	He feels	
0,1,0,2,3,0,1	better, it, sunny	He feels	good, bad, strange	He feels better	
1,0,2,3,0,1	it, sunny	feels better	since, here, too	He feels better since	
0,2,3,0,1	it, sunny	better since	today, that, then	He feels better since it	
2,3,0,1	sunny	since it	was, is, knows	He feels better since it is	
3,0,1	sunny	it is	what, known, a	He feels better since it is a	
0,1	sunny	is a	book, dog, pencil	He feels better since it is a	
				sunny	
1	–	a sunny	day, park, side	He feels better since it is a	
				sunny day	

Experimental results

In order to obtain the results, an English and a multilingual file have been created using Wikipedia data. Each file contains lines in nine different size ranges: 0-99, 100-199, 200-299, …, 800-899. For each range, 100 lines are selected for English and 70 lines consisting of 10 lines from seven languages (English, German, French, Spanish, Italian, Dutch and Turkish) are selected for the multilingual file. In total, there are 900 lines in the English file and 700 lines in the multilingual file. Total file size is 564 KB for English text and 388 KB for multilingual text.

The performance of MLMCompress is compared with the short text compression methods Shoco and Smaz, the general-purpose compression methods Gzip and Zlib, and the learning-based compression methods NNCP and GPTZip. The results are obtained on a computer running Windows 11 with an Intel Xeon Gold 6248R CPU and 256 GB RAM.

Python implementation of MLMCompress is used in the tests. The default implementations available within the Python library have been used for Gzip and Zlib. Source codes used for other methods are obtained from the following sources:

• Python implementation of NNCP v2 is obtained from https://bellard.org/nncp/

• C implementation of Shoco is obtained from https://github.com/Ed-von-Schleck/shoco

• C implementation of Smaz is obtained from https://github.com/antirez/smaz

• Python implementation of GPTZip is obtained from https://github.com/erika-n/GPTzip

The proposed method has been compared with other methods in terms of compression ratio, compression speed, and decompression speed. The compression ratio (CR) is calculated using the compressed file size (Sc) and the uncompressed raw file size (Sr) in bytes, as shown in Eq. (1). (1) CR=Sc∗100/Sr.

To calculate the compression and decompression speeds, the number of bytes processed by an algorithm per second (bytes/s) is measured. The calculation used for compression speed (CS), where the compression time is denoted as tc, is provided in Eq. (2). (2) CS=Sr/tc.

At this stage, since the data processed during compression is raw, the compression speed is calculated based on the number of bytes of raw data processed per second. For the calculation of decompression speed (DS), where the decompression time is denoted as td, is shown in Eq. (3). (3) DS=Sc/td.

In contrast, the decompression speed is calculated based on the compressed data since it processes compressed data in td. The times are measured in seconds, and the results are normalized by the data sizes in bytes, yielding speeds in bytes/s. The obtained results are discussed in the relevant subsections.

Compression ratio results

MLMCompress is tested with different window sizes (n = 5, 10, 15) and different prediction sizes (r = 64, 128, 254). The compression ratio results of the tests performed with four different models on English and Multilingual short texts are given in Table 5.

Table 5 Compression ratios of English and multilingual short texts using different window size and prediction size parameters with MLMCompress (%).

Model & Language	Size	n = 5	n = 5	n = 5	n = 10	n = 10	n = 10	n = 15	n = 15	n = 15	
	Range	r = 64	r = 128	r = 254	r = 64	r = 128	r = 254	r = 64	r = 128	r = 254	
Model A—English	0–99	73.69	68.72	63.98	72.25	67.00	63.01	71.71	66.71	62.79	
100–199	69.43	64.07	59.26	65.16	60.72	56.65	63.88	59.48	55.71	
200–299	67.03	61.22	57.23	62.11	57.59	54.15	60.90	56.66	53.16	
300–399	63.41	56.80	52.86	56.77	52.69	49.09	55.18	51.53	48.00	
400–499	64.80	56.85	52.72	57.48	53.12	49.32	55.69	51.65	47.90	
500–599	63.64	55.90	51.98	56.57	52.45	48.39	54.78	50.93	46.92	
600–699	63.56	55.21	51.18	55.45	51.22	47.73	53.92	49.82	46.25	
700–799	65.35	56.37	52.35	56.69	52.57	48.61	55.08	50.70	47.03	
800–899	63.98	55.59	51.60	56.04	51.66	47.80	54.25	50.27	46.47	
Model B—Multilingual	0–99	80.95	78.70	75.74	78.31	75.49	73.47	78.39	75.36	73.41	
100–199	65.35	62.38	59.95	61.83	59.49	57.09	61.21	58.73	56.83	
200–299	65.97	62.84	60.13	61.87	59.29	56.49	60.93	57.83	55.62	
300–399	67.60	64.15	61.31	64.19	61.10	58.22	63.46	60.06	57.51	
400–499	64.06	60.45	57.52	61.46	58.15	55.19	60.38	57.26	54.51	
500–599	63.84	60.77	57.48	61.26	57.95	55.07	60.38	57.32	54.33	
600–699	63.81	60.62	57.42	60.98	57.83	54.75	59.83	56.72	54.06	
700–799	62.69	59.37	56.28	60.05	56.83	53.98	59.29	56.05	53.19	
800–899	63.50	60.38	57.34	60.93	57.74	54.93	60.34	57.02	53.96	
Model C—English	0–99	95.27	95.75	94.25	96.90	96.08	94.16	96.91	95.86	93.87	
100–199	89.36	93.03	91.15	93.89	92.90	90.93	93.81	92.56	90.52	
200–299	87.37	89.84	88.25	90.88	89.78	88.05	90.76	89.67	87.52	
300–399	84.29	84.48	82.85	85.43	84.36	82.56	85.33	84.13	82.03	
400–499	83.43	84.58	82.80	85.55	84.50	82.63	85.53	84.22	82.13	
500–599	82.27	83.00	81.33	83.99	82.95	81.09	83.91	82.73	80.67	
600–699	81.45	82.19	80.67	83.09	82.16	80.43	83.06	81.92	79.95	
700–799	82.40	84.00	82.24	84.89	83.83	81.99	84.86	83.57	81.54	
800–899	81.34	81.52	79.93	82.44	81.47	79.67	82.42	81.32	79.30	
Model D—Multilingual	0–99	92.60	91.47	90.76	91.62	90.79	90.34	91.48	90.67	90.13	
100–199	82.36	81.45	80.29	81.74	80.37	79.45	81.81	80.56	79.20	
200–299	81.45	80.10	78.67	81.03	79.49	78.31	80.94	79.65	78.15	
300–399	81.72	80.46	78.91	81.27	79.99	78.52	81.12	79.84	78.46	
400–499	79.73	78.12	76.85	79.28	77.70	76.26	79.11	77.68	76.02	
500–599	78.58	77.13	75.73	78.31	76.90	75.48	78.11	76.80	75.39	
600–699	78.35	77.01	75.46	77.98	76.60	74.83	77.80	76.52	74.96	
700–799	77.87	76.48	74.99	77.52	76.17	74.78	77.50	76.01	74.40	
800–899	78.20	76.99	75.62	78.06	76.65	75.17	78.00	76.67	75.34	

As seen in Table 5, most of the times both for English and Multilingual texts, combination of w = 15 and p = 254 gives the best compression results for all different length ranges and models. Therefore, for comparison with other compression algorithms, w = 15 and p = 254 results are used for all models.

The compression ratio results of the English short texts for different methods are given in Table 6. As can be seen in the table, MLMCompress Model A achieved the best compression ratios across all size ranges. NNCP, which is another learning-based method in our test and compresses the 1GB ‘enwik9’ text data at the highest rate today (based on Large Text Compression Benchmark results in November 2023), expands short texts instead of compressing them. Since NNCP default cannot compress most texts larger than 500 bytes and NNCP lstm cannot compress most texts larger than 600 bytes, results regarding these sizes are not given.

Table 6 Compression ratios of English short texts with different methods (%).

Size Range	Gzip	Zlib	Shoco	Smaz	MLMComp Model A	MLMComp Model C	NNCP default	NNCP lstm	NNCP lstm-fast	NNCP enwik8	GPTZip	
0–99	121.98	104.08	73.80	73.60	62.79	93.87	265.15	224.89	222.20	405.40	51.01	
100–199	97.23	86.58	76.26	76.01	55.71	90.52	177.47	158.18	155.78	240.79	41.02	
200–299	82.61	75.25	76.50	76.14	53.16	87.52	146.75	134.46	132.36	182.97	36.24	
300–399	73.71	67.03	74.97	73.58	48.00	82.03	134.00	124.64	122.84	159.11	31.83	
400–499	68.61	64.06	73.55	72.55	47.90	82.13	126.77	119.26	117.73	145.77	29.93	
500–599	65.49	60.83	72.72	71.61	46.92	80.67	–	115.64	114.24	137.17	28.77	
600–699	62.95	59.10	71.77	70.88	46.25	79.95	–	–	112.26	131.64	28.14	
700–799	61.20	57.59	71.58	71.31	47.03	81.54	–	–	110.76	127.70	26.92	
800–899	59.67	56.63	71.35	70.45	46.47	79.30	–	–	109.58	124.26	26.34	

According to the test results, the method with the best compression ratio among the NNCP versions is lstm-fast, and the method with the worst compression ratio is enwik8. While Gzip cannot compress texts smaller than 100 bytes, the compression ratio increases significantly as the text size increases, similar to NNCP. Gzip gave similar results with the short text compression methods Shoco and Smaz in the size range of 300-399 bytes, and started to give better compression rates than them for texts larger than 400 bytes.

The compression ratio results of the tests performed on multilingual short texts are given in Table 7. Similar to the English results, the best compression ratios in every size range for multilingual texts are obtained with MLMCompress that uses a large model (Model B), and the worst compression ratios are obtained with NNCP enwik8. As seen in Tables 6 and 7, MLMCompress achieves worse compression ratio results with tiny models (Model C and Model D). This is because larger models have a higher prediction accuracy, which allows for an increase in the number of compressed words. GPTZip achieves up to 20% better compression than MLMCompress in the best scenarios, this advantage diminishes with multilingual files.

Table 7 Compression ratios of multilingual short texts with different methods (%).

Size Range	Gzip	Zlib	Shoco	Smaz	MLMComp Model B	MLMComp Model D	NNCP default	NNCP lstm	NNCP lstm-fast	NNCP enwik8	GPTZip	
0–99	128.45	105.24	86.88	85.26	73.41	90.13	281.42	237.97	234.74	437.09	77.78	
100–199	97.11	87.16	83.48	81.54	56.83	79.20	179.97	161.04	158.07	247.24	64.94	
200–299	82.81	75.57	85.21	82.34	55.62	78.15	147.26	135.74	133.16	185.18	58.93	
300–399	74.19	67.99	88.01	84.21	57.51	78.46	134.21	125.65	123.45	160.49	54.26	
400–499	68.94	64.76	84.43	80.80	54.51	76.02	126.87	120.01	118.02	146.58	51.33	
500–599	65.78	62.10	84.12	79.98	54.33	75.39	–	116.54	114.79	138.07	49.27	
600–699	63.57	59.85	84.15	80.28	54.06	74.96	–	–	112.71	132.59	47.86	
700–799	61.72	57.77	83.93	79.73	53.19	74.40	–	–	111.09	128.19	47.06	
800–899	59.84	57.36	83.49	79.39	53.96	75.34	–	–	109.87	124.88	46.47	

Gzip and all versions of NNCP gave similar results in both tables regardless of the language of the text. However, Smaz and Shoco methods compressed multilingual texts with a much worse compression ratio than English texts due to their static dictionary structures specific to English. While the compression ratio increases significantly as the size of the texts increases in NNCP and Gzip methods, the effect of text size on the compression ratio is much less in short text compression methods.

In MLMCompress, Model A’s compression ratio for English short texts is higher than Model B’s compression ratio for multilingual short texts. However, the opposite is the case for Model C and Model D. Although the compression ratio improves as text size increases in MLMCompress, the difference is not as much as in NNCP and Gzip.

Compression and decompression speed results

Learning-based compression methods perform compression and decompression much slower than traditional methods. For this reason, the MLMCompress method is compared in this section only with NNCP. Compression and decompression speeds in nine different size ranges are given in Table 8 for English texts and in Table 9 for multilingual texts. In these tables, as in the previous section, w = 15 and p = 254 results are used for all MLMCompress models. This configuration, which offers the highest compression ratio, is actually the slowest in terms of compression and decompression speeds.

Table 8 Compression and decompression speeds of English short texts with MLMCompress, NNCP and GPTZip (bytes/s).

	Size Range	MLMComp Model A	MLMComp Model C	NNCP default	NNCP lstm	NNCP lstm-fast	NNCP enwik8	GPTZip	
Compression speeds	0–99	398.78	2392.67	830.80	715.44	316.04	12.12	58.55	
100–199	365.49	2140.71	1472.08	1267.12	659.75	21.15	57.26	
200–299	397.89	2278.82	2039.63	1759.08	927.27	28.87	57.23	
300–399	475.89	2316.00	2413.94	2153.60	1074.28	34.00	57.31	
400–499	502.75	2237.25	2782.97	2396.23	1372.46	37.11	57.26	
500–599	541.12	2299.75	–	2612.36	1343.61	39.92	57.1	
600–699	553.53	2312.96	–	–	1530.61	41.79	56.86	
700–799	563.85	2255.39	–	–	1672.01	43.47	56.97	
800–899	592.38	2289.46	–	–	1638.59	45.07	60.83	
Decompression speeds	0–99	239.27	7178.00	887.31	783.62	323.77	12.15	58.78	
100–199	202.50	4995.00	1536.29	1375.02	676.40	21.24	57.50	
200–299	203.80	5013.40	2126.56	1865.80	944.32	28.97	57.44	
300–399	203.16	5790.00	2503.95	2255.71	1093.90	33.95	57.46	
400–499	197.99	4971.67	2856.95	2535.29	1407.19	37.17	57.43	
500–599	200.71	5519.40	–	2763.58	1371.41	39.99	57.44	
600–699	203.02	5396.92	–	–	1562.62	41.93	57.11	
700–799	197.42	5316.29	–	–	1701.79	43.43	57.34	
800–899	205.61	5647.33	–	–	1663.10	45.07	61.42	

Table 9 Compression and decompression speeds of multilingual short texts with MLMCompress, NNCP and GPTZip (bytes/s).

	Size Range	MLMComp Model B	MLMComp Model D	NNCP default	NNCP lstm	NNCP lstm-fast	NNCP enwik8	GPTZip	
Compression speeds	0–99	299.57	2296.67	1251.36	993.94	434.89	16.90	57.73	
100–199	281.83	2931.00	2124.21	1789.38	926.94	29.63	57.28	
200–299	297.95	3091.25	3153.94	2553.70	1387.30	41.48	57.11	
300–399	320.79	2689.69	3910.74	3126.15	1550.67	49.10	57.26	
400–499	345.79	2644.29	4337.42	3456.86	1978.56	54.62	57.04	
500–599	363.22	2760.45	–	3798.08	1987.36	58.36	56.82	
600–699	370.92	2689.17	–	–	2186.24	61.69	57.28	
700–799	380.89	2764.96	–	–	2427.85	63.35	56.77	
800–899	378.88	2725.48	–	–	2374.05	65.07	56.78	
Decompression speeds	0–99	168.05	3445.00	1335.27	1091.57	445.90	16.96	58.15	
100–199	164.66	3663.75	2284.49	1936.70	952.86	29.80	57.39	
200–299	163.77	3091.25	3263.38	2709.54	1430.72	41.49	57.37	
300–399	175.71	3496.60	4048.41	3265.72	1578.32	49.14	57.63	
400–499	172.90	3457.92	4488.13	3678.34	2029.03	54.61	57.23	
500–599	179.25	3943.50	–	4006.75	2036.41	58.42	57.16	
600–699	178.78	3796.47	–	–	2231.83	61.67	57.54	
700–799	181.64	3929.16	–	–	2472.56	63.36	56.92	
800–899	182.48	4224.50	–	–	2420.71	65.15	57.13	

While NNCP’s compression and decompression speeds increase as text size increases, in MLMCompress text size has little effect on compression and decompression speeds. As seen in Tables 8 and 9, NNCP’s compression and decompression speeds are very close to each other in all versions.

In MLMCompress, compression is faster when large models are used and decompression is faster when small models are used. The reason why decompression is faster in smaller models is that less decompression is done with BERT since fewer words can be compressed. MLMCompress Model A is 4–6 times slower in compression and 25–30 times slower in decompression than MLMCompress Model C. Similarly, Model B is 7–10 times slower than Model D in compression and 19-23 times slower in decompression.

The unexpected result about NNCP is that the lstm-fast version performs compression and decompression operations about two times slower than lstm. Since the compression ratio of lstm-fast is also better than lstm, probably the lstm and lstm-fast parameters may have been assigned inversely in the Python library we use. The fastest NNCP version, default, is approximately 16% faster in compression and approximately 25% faster in decompression than lstm. NNCP enwik8, which gives the worst results in compression ratio, is also very slow compared to other methods in compression and decompression.

As noted above, although GPTZip can produce better results than MLMCompress, these differences become significantly more pronounced when considering the speed difference. In this context, MLMCompress demonstrates a balanced performance in terms of both compression ratio and speed. This balance can be attributed to the size of the model and the number of parameters used.

To ensure the accuracy of the results, sentences of varying lengths are extracted from the LAReQA dataset (Roy et al., 2020) using a step size of 100, similar to the previous dataset. The compression ratio, compression speed, and decompression speed results for these sentences are calculated. The results are presented in Table 10.

Upon examining the data presented in Table 10, it is evident that GPTZip can produce better compression results than MLMCompress. However, when the speeds at which these results are achieved are considered, the difference becomes strikingly significant. For instance, while producing faster results, GPTZip compresses files 6.3% more effectively than MLMCompress in its best case, yet remains 10 times slower in terms of speed.

Discussion

In this paper, we propose MLMCompress, a method that utilizes the transformer architecture for word prediction. The core idea of this method is to leverage transformers to bypass obtaining word frequency and occurrence probability information during the first pass through the file, as in well-known semi-static compression methods. This approach eliminates the need for another pass through the file. Additionally, it is evident that the prediction accuracy of a transformer model surpasses the statistical estimation derived from a limited dataset in a file-based approach. As the performance of the model used increases, the compression efficiency of MLMCompress will also improve.

Table 10 Average ratio and speed results for dataset created from Roy et al. (2020).

Method	Compression ratio	Compression speed (B/s)	Decompression speed (B/s)	
Gzip	74.94	3.13 × 1015	2.08 × 1016	
Zlib	69.18	1.19 × 107	1.76 × 107	
Shoco	72.29	1.43 × 107	5.70 × 107	
Smaz	77.60	1.51 × 107	3.25 × 107	
MLMCompress Model A	63.23	904.85	822.12	
MLMCompress Model B	56.54	567.98	551.45	
MLMCompress Model C	67.91	622.19	1879.88	
MLMCompress Model D	68.71	973.99	4450.46	
NNCP default	153.00	8721.21	8900.25	
NNCP lstm	133.52	5342.49	7047.79	
NNCP lstm-fast	125.59	2190.17	2217.52	
NNCP enwik8	165.54	51.12	51.58	
GPTZip	50.28	61.85	62.10	

Another advantage of MLMCompress is its ability to work with any model capable of making word predictions. With the ongoing development and availability of language models, customization has become significantly easier, and various models can be trained and deployed. In scenarios where compression is performed on similar data, a small model can be trained over many epochs to achieve fast and high compression ratios. Alternatively, models can be developed for different scenarios or languages, which can then be used with MLMCompress to create domain-specific compression methods.

One limitation of using MLMCompress is that the model must be present on both the compression and decompression sides. When using large, general models as in this study, this disadvantage may be mitigated in certain cases since these models are publicly available and widely applicable. In situations where small models are trained for specific tasks, and the method is expected to be used repeatedly, the storage requirement for these models might be negligible. However, if very large customized models are to be used, this advantage diminishes.

Another limitation, similar to NNCP, is the resource and speed requirements inherent to model-based methods. To address the speed disadvantage, MLMCompress reduces the overhead by batching requests for the next word prediction, minimizing the frequency of model invocations. Nevertheless, as the models grow larger, the required resources increase, making it clear that configurations capable of running large language models will be necessary.

Given these constraints, MLMCompress is most advantageous when used for compressing short texts in situations where data diversity is low and usage is frequent. For this reason, short texts are used in the experiments, and as shown in the experimental results section, the method outperformed traditional approaches.

Conclusion

The widespread availability and versatility of transformer models have enabled their use in compression tasks. While there are probability-based AI methods for compression, it is also possible to use the text generated by language models. Although these methods could be employed for compression as their word prediction accuracy improves, they are not suitable for compressing large texts due to their very slow processing times. Based on the idea that these methods are more appropriate for short text compression, this study focuses on short text compression and introduces a word-based compression method using BERT, one of the most popular transformer architectures.

In this study, we propose MLMCompress, a compression method that leverages transformer architectures for word-based prediction. MLMCompress can perform compression using any model capable of word prediction, making it highly customizable for different languages, speed, and storage requirements.

MLMCompress is compared with other transformer-based compression methods, NNCP and GPTZip, the most widely used general-purpose compression methods, Gzip and Zlib, and the short-text compression methods, Smaz and Shoco. In tests conducted on English and multilingual short texts, MLMCompress, when using large models (Model A and B), produced the best compression ratio results, outperforming NNCP by 38% for English and 42% for multilingual texts. NNCP, in contrast, yielded the worst results. Furthermore, MLMCompress demonstrated up to 180% faster compression and decompression speeds compared to NNCP.

Although both NNCP and MLMCompress employ similar compression logic, the significant difference in compression ratio is due to the additional information that NNCP stores alongside the compressed data. To achieve a good compression ratio on short texts with MLMCompress, static Huffman code tables are used instead of storing the Huffman table data along with the compressed text.

If MLMCompress is used with small models, the compression ratio will be lower, but compression and decompression can be performed much faster. When used with small models (Models C and D), MLMCompress can compress and decompress faster than NNCP (especially for the shortest texts) while still offering significantly better compression ratios.

Although the size of the BERT model incurs a cost in terms of compression, this cost can be disregarded in scenarios where the model, like the one used in this study, is well-known and standardized or can be provided online, thereby eliminating the need for downloading the model onto each individual computer.

Considering that MLMCompress can be used with any word prediction model, future research aims to achieve higher compression by using larger models, accepting the trade-off in speed, or by training smaller, specialized language models that can outperform transformer models without significant speed loss. This approach would allow the method to perform efficiently in specific scenarios and be tailored to achieve the best results for specific datasets. Additionally, there is a plan to develop a version that increases the compression ratio by employing a variable-length window, even at the cost of longer compression times, by foregoing the batch request mechanism used in the current model.

Supplemental Information

Supplemental Information 1 Dataset containing text in 7 different languages for obtaining experimental results

Each line with different length in raw data for given dataset is used for obtaining compression ratio, compression and decompression times for different languages and algorithms.

Supplemental Information 2 Raw data used for finetuning english models.

Supplemental Information 3 Raw data used for finetuning multilingual model.

Supplemental Information 4 Test code for obtaining the experimental results of the article

The code file can be run using the models at Github: github.com/emirozturk/MLMCompress.

Additional Information and Declarations

Competing Interests

Author Contributions

Data Availability

The authors declare there are no competing interests.

Emir Öztürk conceived and designed the experiments, performed the experiments, analyzed the data, performed the computation work, prepared figures and/or tables, authored or reviewed drafts of the article, and approved the final draft.

Altan Mesut conceived and designed the experiments, performed the experiments, analyzed the data, performed the computation work, prepared figures and/or tables, authored or reviewed drafts of the article, and approved the final draft.

The following information was supplied regarding data availability:

The data, model and code is available at GitHub and Zenodo:

- https://github.com/emirozturk/MLMCompress.

- Emir Öztürk. (2024). emirozturk/MLMCompress: First Release (1.0.0). Zenodo. https://doi.org/10.5281/zenodo.13629551.

The data used for experiments and fine-tuning operations are available in the Supplemental Files.

The original english50mb file, from which a section is taken, is available at Pizza&Chili Corpus: https://pizzachili.dcc.uchile.cl/texts.html.

The original datasets used for creating multilingual texts for experiments is available at https://huggingface.co/datasets (Terms: GermanQuAD, Spanish speech text, xnli2.0 train French, Italian tweets 500k, dutch social and turkish186).

The original wikipedia data for different languages from which a section is taken, is available at: https://dumps.wikimedia.org.

The multilingual dataset is available at GitHub: https://github.com/google-research-datasets/lareqa.

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
