# Peer review of "Learning-based short text compression using BERT models"

_PeerJ Computer Science, doi:10.7717/peerj-cs.2423_

## Round 0.1 · original submission · Major Revisions

Thank you for submitting your manuscript to PeerJ Computer Science. The review process has been completed, and we have carefully considered the feedback provided by the reviewers.

The reviewers have acknowledged the potential value of your work but have raised several significant concerns, particularly regarding the methodology and experimental evaluation. These concerns require substantial revisions to ensure that the manuscript meets the rigorous standards of our journal.

In light of these comments, I am recommending that your manuscript undergoes a major revision. We encourage you to carefully address each of the reviewers’ comments, paying close attention to the methodological issues and the robustness of your experimental evaluation. A detailed response to the reviewers, explaining the changes made or providing justifications for any unaddressed points, should accompany your revised submission.

Once the revisions have been completed, your manuscript will undergo a further round of review to ensure that all major concerns have been satisfactorily addressed.

We appreciate the effort that you have put into this research and look forward to receiving your revised manuscript.

Reviewer 1 ·

Basic reporting

1- Elaborate on the features of the dataset, its total size, as well as the size of training and testing sets, present this information in a tabular form for easy comprehension.
2- Address grammatical errors and typographical mistakes through proofreading.

Experimental design

1- Include a flowchart detailing the steps of the algorithm.
2- Describe the architecture of the proposed model, ensuring it is comprehensible.
3- Provide all the parameters used in the analysis in a clear table.
4- Conduct a comparative evaluation of the proposed model's performance against other current leading methods using the same dataset.
5- The time spent on the experiments should be recorded and shared.
6- State the resolution of the figures used to ensure they are clear and easy to understand.

Validity of the findings

1- All metrics and measurements used should be calculated in the experimental results.
2- Mention the improvement percentages in the level of accuracy and highlight the significance of these results in the abstract and conclusion sections.
3- Include a detailed Limitation and Discussion section.

Additional comments

1- Include potential future work and recommendations in the conclusion sections.
2- Also, a high-resolution insert for figures and flowchart should be added for clarity.

Cite this review as

Reviewer 2 ·

Basic reporting

The English language needs some improvement and proper proofreading for the abstract section and in explaining the abbvation like NNC Cmix
Ex: MLMCompress is not proper introduced, and the approach used in not explicitly mentioned “using four different models, two of which are tiny and fine-tuned”.
The introduction section lacks a clear definition of the problem and an explanation of the actual BERT model. Also, more related works are needed with explain the difference between this work and other works. Conclusion needs to be more summary about the approaches used and the results achieved. usually, conclusion is two long paragraphs

Experimental design

The methodology section lacks some information about the dataset used, evaluation equation, fine-tuning parameters used, and other. It is also not clearly explained as it is not reproducible by another investigator.

This approach to compression using and Huffman coding represents a sophisticated integration of modern language models with traditional compression techniques.
Strengths
Integrating BERT predictions with Huffman coding creates a hybrid compression method that combines the strengths of deep learning with traditional compression, potentially offering better performance compared to using either technique in isolation. The approach effectively utilizes a powerful language model like BERT to predict the next word in a sequence, which can potentially lead to high compression ratios.

Limitations
The compression effectiveness is heavily reliant on BERT's prediction accuracy. If BERT fails to predict the next word, the compression ratio suffers due to the need to store the word in the R array. Also, the use of a fixed window size might limit the model's ability to capture long-range dependencies in the text. A variable window size could potentially improve performance. Moreover, the provided example is a small-scale demonstration. A comprehensive evaluation on diverse text corpora is necessary to assess the approach's overall effectiveness.
Some of these Limitations need to be addressed in Threats to validity section of in conclusion

Validity of the findings

no comment'

Additional comments

no comment

Cite this review as

---

## Round 0.2 · accepted · Accept

After carefully reviewing the revisions you have made in response to the reviewers' comments, I am pleased to inform you that your manuscript has been accepted for publication in PeerJ Computer Science.

Your efforts to address the reviewers’ suggestions have significantly improved the quality and clarity of the manuscript. The changes you implemented have successfully resolved the concerns raised, and the content now meets the high standards of the journal.

Thank you for your commitment to enhancing the paper. I look forward to seeing the final.

Reviewer 1 ·

Basic reporting

- Accept.

Experimental design

-

Validity of the findings

-

Additional comments

-

Cite this review as